# Curvature sensing lipid dynamics in a mitochondrial inner membrane model

Vinaya Kumar Golla [1,2], Kevin J. Boyd[1,3] & Eric R. May [1✉]

Membrane curvature is essential for many cellular structures and processes, and factors such as leaflet asymmetry, lipid composition, and proteins all play important roles. Cardiolipin is the signature lipid of mitochondrial membranes and is essential for maintaining the highly curved shapes of the inner mitochondrial membrane (IMM) and the spatial arrangement of membrane proteins. In this study, we investigate the partitioning behavior of various lipids present in the IMM using coarse-grained molecular dynamics simulations. This study explores curved bilayer systems containing phosphatidylcholine (PC), phosphatidylethanolamine (PE), and cardiolipin (CDL) in binary and ternary component mixtures. Curvature properties such as mean and Gaussian curvatures, as well as the distribution of lipids into the various curved regions of the cristae models, are quantified. Overall, this work represents an advance beyond previous studies on lipid curvature sensing by simulating these systems in a geometry that has the morphological features and scales of curvature consistent with regions of the IMM. We find that CDL has a stronger preference for accumulating in regions of negative curvature than PE lipids, in agreement with previous results. Furthermore, we find lipid partitioning propensity is dominated by sensitivity to mean curvature, while there is a weaker correlation with Gaussian curvature.

[1] Department of Molecular and Cell Biology, University of Connecticut, Storrs, CT 06269, USA. [2] Present address: Department of Cell Biology, University of Virginia School of Medicine, Charlottesville, VA 22903, USA. [3] Present address: NVIDIA, 2860 County Hwy G4, Santa Clara, CA 95051, USA. ✉email: eric.may@uconn.edu

Mitochondria are termed the powerhouses of the cell, and unlike other cellular organelles, mitochondria contain a two-layered envelope consisting of the inner (IMM) and outer mitochondrial membrane (OMM)[1]. The structure of the OMM is similar to the outer cell membrane and consists of various pore-forming proteins, allowing the exchange of material into and out of intermembrane space (IMS). The IMM separates the IMS from the mitochondrial matrix, and the structure of the IMM includes highly curved invaginations called cristae, which house various proteins involved in the electron transport chain[2]. The dimensions of the cristae can vary in length, width, and geometry of the opening (cylindrical vs. ellipsoidal), with the smallest opening diameters being in the range of 20 nm[3]. The morphology of the IMM is maintained by several proteins such as the mitochondrial contact site and cristae organizing system (MICOS) complex, ATP synthase, and mitofilin[4]. At the same time, the lipid composition of the mitochondrial membrane plays a vital role in the cristae morphology, and the membrane stress imposed by the degree of curvature can be reduced by the presence and enrichment of non-bilayer forming phospholipids. Phosphatidylcholine (PC), phosphatidylethanolamine (PE), and cardiolipin (CDL) lipids are among the most abundant mitochondrial membrane lipids. Both PC and PE lipids account for 80% of total membrane lipids, and CDL lipids are measured up to 20%[1,5].

The phospholipid CDL is a unique lipid and is primarily found in mitochondrial membranes in eukaryotes[1]. CDL is mainly located in the IMM and associated with many mitochondrial functions, such as maintaining membrane morphology, partaking in mitochondrial signaling pathways, and aiding in the proper insertion and folding of membrane proteins involved in the electron transport chain reactions[2,6,7]. Additionally, CDL regulates membrane fluidity, and thus, the presence or absence of CDL systematically modifies the membrane properties, which can affect the insertion and rotation of membrane proteins and biomolecules into the mitochondrial membranes and thereby impact mitochondrial function[8]. Defects in the CDL biosynthetic pathways are associated with many mitochondrial dysfunctions and cardiac diseases, and alteration in the concentrations of CDL subtypes results in changes in the oxidative phosphorylation process, leading to cellular defects[9]. Similarly, modifications of CDL biogenesis and remodeling pathway, such as the mutation in the TAZ gene, causes Barth syndrome[10,11]. Barth syndrome is a rare cardiomyopathy associated with skeletal muscle weakness, neutropenia, and growth retardation[12].

PE and CDL are considered to be in the class of non-bilayer forming lipids and are shown to be essential for mitochondrial structure and function[13]. Due to their conical molecular geometries, PE and CDL have a propensity to aggregate into non-bilayer structures such as inverse hexagonal ($H_{II}$) phases[14]. It is assumed that the presence of CDL and PE lipids are essential for cristae formation[15]. Both PE and CDL are partitioned to the negatively curved monolayer facing the cristae lumen. Subsequently, the opposing positively curved monolayer, i.e., mitochondrial matrix facing, contains a large number of PC lipids[4]. Understanding the partitioning, localization, and curvature properties of these lipids in various bilayer compositions in mitochondrial systems is quite complex and challenging.

Numerous atomistic and coarse-grained (CG) simulations studies have contributed to characterizing the properties of cardiolipin, primarily in flat bilayer systems[5,16–21]. More recently, the phenomena of curvature-based lipid sorting have been examined in CG simulations of bilayer nanotubes, and buckled membranes[20,22–24]. Similarly, other studies reported the sorting of lipids into curved regions using in-vitro experiments under various conditions[25–29]. A potential drawback in previous simulation studies examining curvature effects is that the high degree of enforced curvature does not reflect biological curvature scales, and observations in those studies may not be relevant to systems with more modest degrees of curvature.

In this work, we performed CG simulations of an IMM model with geometrical features on the length scale of biological cristea (Fig. 1A, B). Another important feature of our IMM model is that it segregates the system into two separate aqueous compartments, analogous to the mitochondrial matrix and IMS. We envision the region on the inside of the cylinder as the IMS and the region outside the cylinder as the matrix. While we do not exploit this feature in the current study, in future work we could examine the effect of transmembrane potentials by varying the ionic species or concentrations in the different solvent compartments. The primary goal of this study was to provide insights into the spatial arrangement of lipids in the IMM model. This model allows for observation of the localization of lipids into the various curved region of the IMM model and the factors that influence the sorting of the lipids. Furthermore, the effect of mean and Gaussian curvature (Fig. 1D) on the segregation of lipids was quantified in various systems containing two or three different lipid species. The lipids that were considered were POPC, POPE, DOPE, and CDL (see Fig. S1 for chemical structures and CG representations). The head group of CDL contains two phosphate moieties, and the ionization state under physiological conditions has been debated. Some studies have support disparate $pK_a$s of the phosphates ($pK_1 \sim 2-4$ and $pK_2 > 8$) while other studies support both phosphates having low $pK_a$s ($pK_1 \approx pK_2 < 2$)[30,31]. We also explored the CDL head group charge effect on lipid partitioning by performing simulations with CDL modeled as a monoanion ($CDL^{-1}$) and as a dianion ($CDL^{-2}$). A summary of simulated systems and compositions is listed in Table S1.

Our findings demonstrate that the intrinsic negative curvature of PE and CDL localizes them in the higher negative curvature regions. In systems containing both CDL and PE lipids, we find that CDL accumulates to a greater extent than PE in negatively curved regions, though the amount of accumulation is sensitive to both the degree of acyl chain saturation on PE and head group charge on CDL. We quantitatively determined the enrichment or depletion of PE, PC and CDL lipids in the various regions of the IMM model. Additionally, we performed a correlation analysis which shows the lipids are very sensitive to the mean curvature of the leaflet and have a modest but significant sensitivity to Gaussian curvature.

## Results

In this study, we investigated the dynamics of lipids in a curved bilayer system, representative of the geometry of an IMM cristae. In order to generate and maintain the geometry of this system, we rely on the Building Unique Membranes in Python (BUMPy)[32] tool and employ dummy particles to prevent shape distortions. Despite the advantage of using dummy particles as shown previously[32–34], a thorough analysis of potential artifacts induced on membrane structure is lacking. Therefore, we ran simulations of flat bilayers proximal to a dummy particle layer with increasing interaction intensity to discern the strength of interaction that induces aberrant lipid structure and dynamics. This protocol has the added benefit of validating our dummy particle force measurement scheme and establishing the potential to utilize them as spatially dependent force sensors to identify where stresses reside in a complex bilayer system. Therefore, we first investigated the effects related to the use of dummy particles, and then, having identified a parameter set where lipid dynamics would be minimally perturbed, the dummy protocol was implemented to study

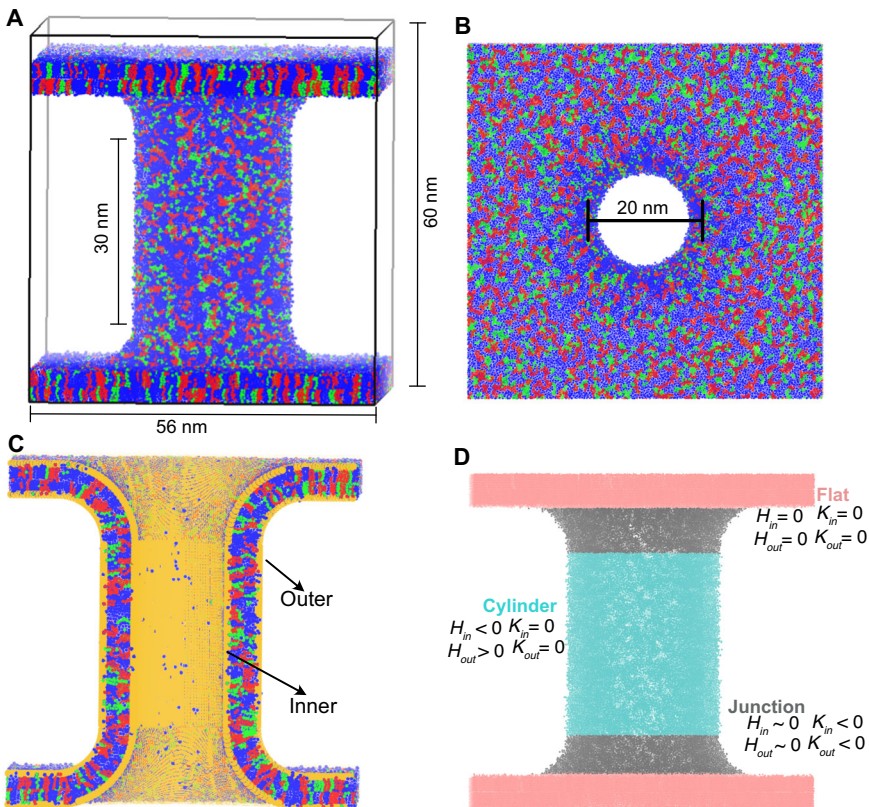

**Fig. 1 The geometry of IMM model systems. A** Side view of a POPC (blue)-POPE (green)-CDL (red) system. **B** Top view of the same system in (**A**). **C** Cutaway view including dummy particles (orange). The inner and outer leaflets are labeled. **D** Image showing the system categorized into three regions of different curvature features: flat (pink), cylinder (cyan), and junction (gray). The sign of the mean ($H$) and Gaussian ($K$) curvatures for each leaflet in each region are indicated.

the IMM model with different lipid compositions and different geometries.

**Effect of dummy particles on lipid and membrane properties.** To evaluate if dummy particle interactions with lipid bilayers induced artifacts in the bialyer properties, several membrane properties were analyzed in a flat POPC/DOPE bilayer system. In this system a single layer of dummy particles was placed above the upper leaflet (see Fig. S2) to evaluate the effect of dummy particle induced forces and other simulation parameter choices, most notably constant pressure (NPT) vs. constant volume (NVT) ensemble simulations. During NPT simulations, the dummy particles are positionally restrained, whereas, during the NVT simulations, the dummy particle positions are frozen. Constant force simulations were run to push the bilayer center of mass (COM) into the wall of dummy particles under the following applied force ($F_{app}$) conditions: $F_{app} =$ 0,1,5,10,20,50,100,250,500, and 1000 kJ mol$^{-1}$ nm$^{-1}$. The bilayer thickness is shown in Fig. S3A, and it can be seen that the membrane thickness increases with the increasing applied constant force for the NPT simulations. However, when examining the range from $F_{app} = 0 - 100$ kJ mol$^{-1}$ nm$^{-1}$ (Fig. S3A inset) a much smaller increase in the membrane thickness is observed, (e.g., 0.22 Å increase when $F_{app} = 20$ kJ mol$^{-1}$ nm$^{-1}$). In the NVT simulations, as expected, no changes in the bilayer thicknesses were observed since the box area cannot fluctuate. Given the relative incompressibility of the bilayer, the area per lipid (APL) shows the opposite trend of the thickness and decreases with the increasing applied forces in the NPT simulations (Fig. S3B). Analogous to the thickness data the perturbations to the APL at low force constants are minimal (e.g. APL = 65.12 Å$^2$

at $F_{app} = 0$ kJ mol$^{-1}$ nm$^{-1}$; APL = 64.58 Å$^2$ at $F_{app} = 20$ kJ mol$^{-1}$ nm$^{-1}$). The observed increase in thickness of the bilayer as the applied force increases is somewhat counter-intuitive. We believe this occurs because as the force increases, the dummy layer exerts a greater repulsive force onto the acyl chains, causing them to extend and straighten and thereby increasing the thickness. This rationale is supported by the increased ordered parameters in the acyl chains at high $F_{app}$ (Fig. S4).

Other lipid properties analyzed include the lipid order parameter (Fig. S4) and tail splay angle (Fig. S5). The second-rank order parameter ($P_2 = \frac{1}{2}(3 \cos^2 \theta - 1)$) was estimated for the consecutive bonds in the CG lipid model[35], where $\theta$ defines the angle between the bond vector and the membrane normal. In NPT, at low applied forces ($F_{app} \leq 250$ kJ mol$^{-1}$ nm$^{-1}$) the order parameter profiles are typical for CG Martini lipids[35] and do not substantially deviate from the equilibrium condition ($F_{app} = 0$), as shown in Fig. S4A, B. We observe increased ordering along the tail at high applied forces, which is consistent with the observed increase in bilayer thickness at a high force constant (Fig. S3A). Under NVT, the order parameters are not significantly affected at any value of applied force tested (Fig. S4C, D). The splay angle was calculated from the angle between the vector connecting the central glycerol bead with the terminal acyl chain bead of the *sn-1* chain and the membrane normal ($z$-direction), and the data is shown in Fig. S5. In NPT, a decrease in the splay angle was observed for the top and bottom layers with increasing force constant. However, a larger splay angle was observed in the top monolayer at the largest applied force, and under this condition, it was observed that the upper monolayer was highly deformed. Up to an applied force of 50 kJ mol$^{-1}$ nm$^{-1}$, the changes in the splay angle were found to be minimally perturbed from

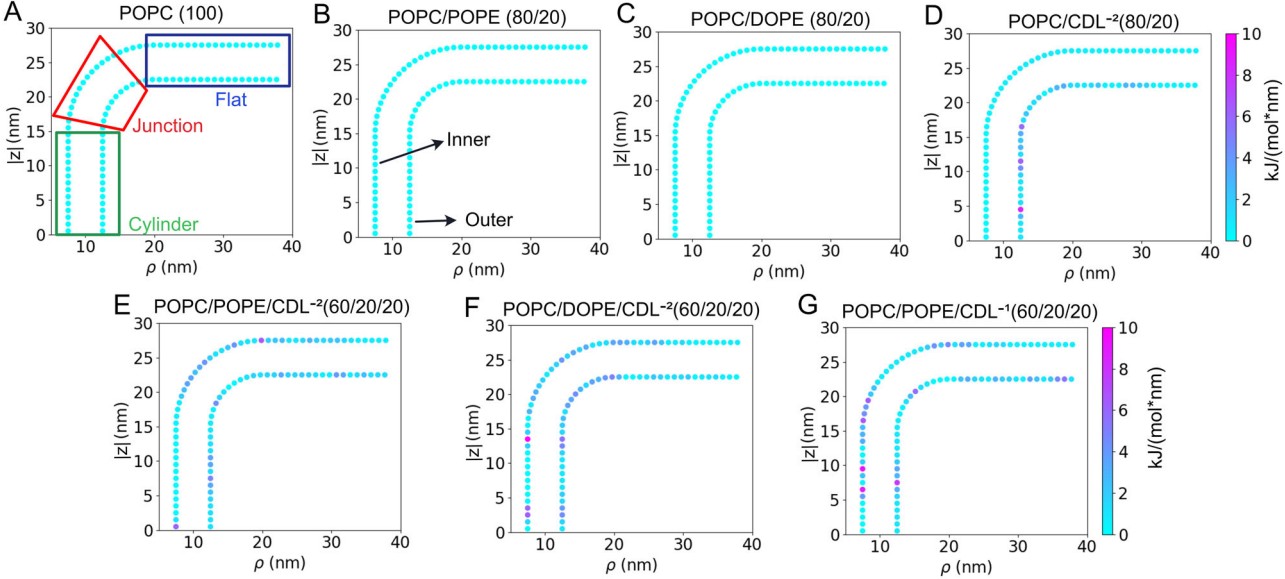

**Fig. 2 Dummy particle forces on IMM systems with $r_{cyl}$ = 10 nm.** System labels match those in Table S1. **A** POPC, (**B**) POPC/POPE, (**C**) POPC/DOPE, (**D**) POPC/CDL$^{-2}$, (**E**) POPC/POPE/CDL$^{-2}$, (**F**) POPC/DOPE/CDL$^{-2}$, and (**G**) POPC/POPE/CDL$^{-1}$. In (**A**) the mapping of the different regions is indicated by the boxed regions, and the inner and outer monolayers are labeled in (**B**) these labels apply to all systems. The color scaling is the same in all figures.

equilibrium. In NVT simulations no significant deviation in the splay angles was observed for any applied force. Overall, the bilayer and lipid property analyses (Figs. S3–S5) indicate that the use of NVT or NPT with dummy particles in a regime where low forces are induced ($F_{app} \leq 50$ kJ mol$^{-1}$ nm$^{-1}$) should not generate significant deviations from native (i.e., equilibrium) lipid structural and dynamic properties and should provide a reasonable approach to study lipid dynamics while controlling bilayer geometry and curvature.

**Dummy particles as force sensors.** In addition to controlling the shape of the bilayer assemblies, we were interested to explore if dummy particles could be used as spatially sensitive force reporters/sensors. As previously described, a series of constant force simulations were conducted in a flat bilayer system under both NPT and NVT conditions. For the NPT simulations, the forces were calculated using an in-house Python script according to $F_i^n = -k_{pr}(a_i^n - A_i^n)$, where $F$ is the force on the $i^{th}$ dummy bead, $n$ is the axis ($x$, $y$ or $z$), $k_{pr}$ denotes the applied positional restraint (1000 kJ mol$^{-1}$ nm$^{-2}$), $a_i$ is the current position of the $i^{th}$ dummy bead, and $A_i$ indicates the reference position of the $i^{th}$ dummy bead. In the case of NVT, the dummy particles are frozen, and the average forces on the dummy particles could be extracted directly from the saved forces in the trajectory files using the *gmx traj* tool. The measured average forces along the $x$, $y$, and $z$ axis are shown in Fig. S6. Since the applied force is only being applied in the $z$-direction, it is sensible that the $x$ and $y$ forces on the dummy particles are approximately equal to zero, and nearly all force is directed along the applied ($z$) direction. It was observed that the calculated average force on the dummy particles in the $z$-direction is equivalent to the applied force (Fig. S6), which provides validation of our ability to measure forces on the dummy particles accurately. Therefore, we can measure naturally occurring forces using the dummy particles, and if the forces are relatively low, we do not expect bilayer properties to be significantly perturbed. To be specific, no significant changes in the bilayer properties were observed at applied forces up to 50 kJ mol$^{-1}$ nm$^{-1}$ and minor changes can be seen at 100 kJ mol$^{-1}$ nm$^{-1}$ (Figs. S3–S5). In the subsequent sections, dummy particles are used to restrain bilayer

systems to mimic a portion of the IMM, where the simulations are performed in the NVT ensemble.

**IMM model force analysis.** As shown in Fig. 1, the IMM model consists of inner and outer monolayers, and each monolayer has a cylinder (Cyl) region, two junction regions (Junc), and two flat regions. To analyze the simulation results in a simplified coordinate system, the three-dimensional (3D) coordinate system ($x$, $y$, $z$) was transformed into a two-dimensional (2D) polar coordinate system ($\rho$, $z$), where $\rho = \sqrt{x^2 + y^2}$. (see Fig. S7 and Fig. 2A). For the seven IMM systems with a 10 nm cylinder radius ($r_{cyl}$) listed in Table S1 (labeled A-G), the average forces on the dummy particles are calculated during the final 1 $\mu$s and are shown in Fig. 2. The first observation is that the average forces remain low in all systems (<10 kJ mol$^{-1}$ nm$^{-1}$). Based on our analysis of lipid and bilayer properties (Figs. S3–S5), the estimated low forces indicate that the dummy particles are not significantly perturbing the native lipid/bilayer properties. The pure and binary systems containing PC and PE lipids (Fig. 2A–C) show uniform stress distributions. Interestingly, systems containing CDL (Fig. 2D–G) have non-uniform stress profiles, and the regions of higher stress are varied from system to system. A consistency in the CDL systems appears to be increased stress in the outer leaflet cylinder which is a region of positive curvature. Given that PE and CDL lipids both have inverted conical molecular geometries (small head groups relative to tail volume)[22,27,36], it is surprising that they give rise to different stress profiles and that the stress is accumulated in regions of positive curvature. The lateral pressure profiles were estimated as a function of the z-direction (along the long cylindrical axis), and are presented in Fig. S8. We observe all systems have low lateral stress in the cylindrical regions but the CDL containing systems have more significant peaks in the flat regions. The normal component of pressure is relatively constant in all systems, which is an indication of mechanical equilibrium. These observations will be further explored in the subsequent sections, which calculate the curvatures in different regions of the IMM and analyze the sorting of various lipids into the different curvature regions.

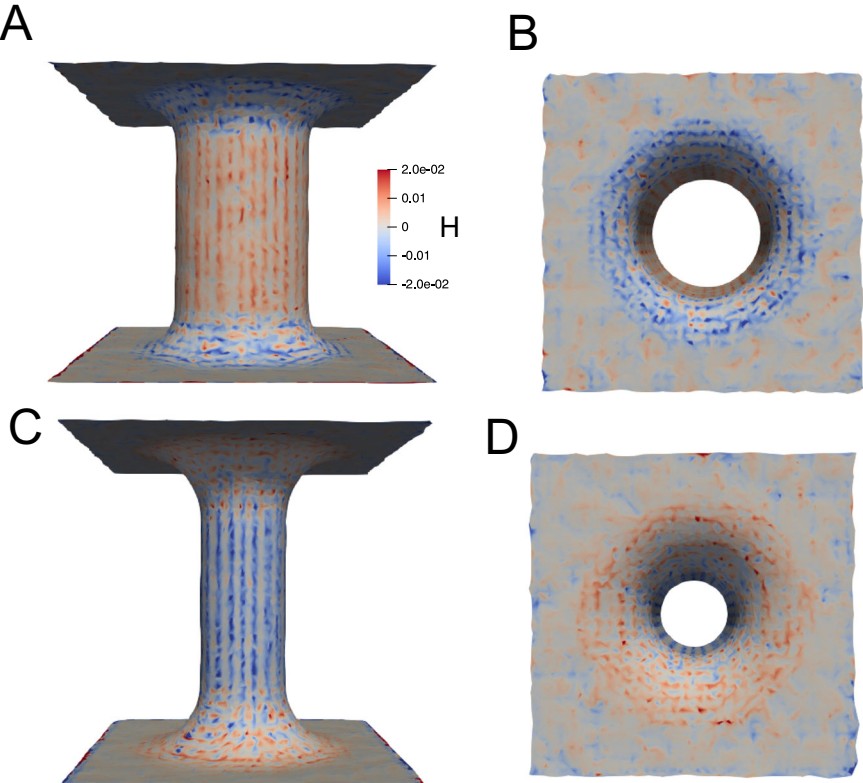

**Fig. 3 Mean curvature of IMM leaflets. A** Side view of IMM outer leaflet, (**B**) top view of IMM outer leaflet, (**C**) side view of IMM inner leaflet, and (**D**) top view of IMM inner leaflet. The color scale bar in (**A**) applies to all subfigures.

**IMM model curvature analysis**. In this section, we calculate the curvature properties of the studied IMM systems. The MemSurfer program[37] was used to compute the mean ($H = \frac{1}{2}(C_1 + C_2)$) and Gaussian curvatures ($K = C_1 C_2$), where $C_1$ and $C_2$ are the surface principal curvatures, for each leaflet of the IMM models. An example of the mean curvature mapped onto the IMM leaflets is shown in Fig. 3. It can be observed that the outer leaflet (Fig. 3A, B) displays primarily positive curvature in the cylinder and primarily negative curvature in the junction, whereas the inner leaflet (Fig. 3C, D) has the opposite signs of curvature, with negative curvature in the cylinder and positive curvature in the junction. As expected, both leaflets display close to zero curvature in the flat regions. Using this analysis method, we can compute the average curvature properties during the CG-MD simulations and then correlate leaflet curvature and lipid segregation properties. It has been reported that the shape of the membrane alone could drive the segregation of lipids without involving any protein-induced bending mechanism[27], and these simulations will allow us to observe this phenomenon in a model with biologically relevant curvature scales.

The mean and Gaussian curvatures were calculated and averaged for the different leaflets and regions during the final 500 ns of the simulations and reported in Table S2. There is little variation between the systems (with the exception of the systems with a differing cylinder radius), as the dummy particles serve to restrain all systems to a consistent geometry. The cylinders display negative mean curvature in the inner leaflets and positive mean curvature in the outer leaflet. The junctions display negative mean curvature in the outer leaflet and near zero mean curvature in the inner leaflet, likely due to principal curvatures of nearly equal and opposite signs. The flat regions displayed negative curvature in the outer leaflet and slightly positive curvature in the inner leaflet. To our knowledge there is limited information on

the relationship between lipid sorting and Gaussian curvature properties. Our system setup allows us to simulate non-uniform and non-vanishing Gaussian curvature regions because the junctions have principal curvatures with opposite signs. We observe that the flat and cylinder regions have near zero $K$ values, while the junctions all have negative $K$ values, with the outer leaflet having a slightly larger magnitude.

**IMM model lipid partitioning**. In this section, we examine the partitioning of different lipids to different regions of the IMM model under varying bilayer compositions (Table S1). To this end, we performed multiple analyses to understand these effects for the systems with $r_{cyl} = 10$ nm. Figures 4–5 displays the average partitioning of lipids determined from the final 1 $\mu$s of the 4 $\mu$s trajectories of studied IMM systems. Through the same coordinate mapping, as shown in the force analysis (Figs. 2 and S7), we compute the lipid composition throughout the IMM and display that data in terms of the overall percentage change (e.g, if a minor component lipid with a bulk percentage of 20% increased to 25% in a localized the region, this would represent a 5% increase). The test system (Fig. 4A) displays good convergence, and nearly the entire system has <1.5% change. Therefore we set 1.5% as the threshold above which we could consider changes as being statistically significant. In all graphs in Figs. 4–5 any localized concentration change below 1.5% was set to zero. In comparing the binary systems (Fig. 4B–D), we see that the partitioning propensity of $CDL^{-2}$ is stronger than either POPE or DOPE. Comparing DOPE vs. POPE, it appears the tail composition has a minimal effect on the partioning propensity, with POPE appearing to have slightly larger accumulation. The maximum localized accumulation of POPE in the inner leaflet is 4.0%, while for DOPE, it is 3.4%. Similarly, the outer leaflet has a maximum localized accumulation of 3.2% for POPE and 3.0% for

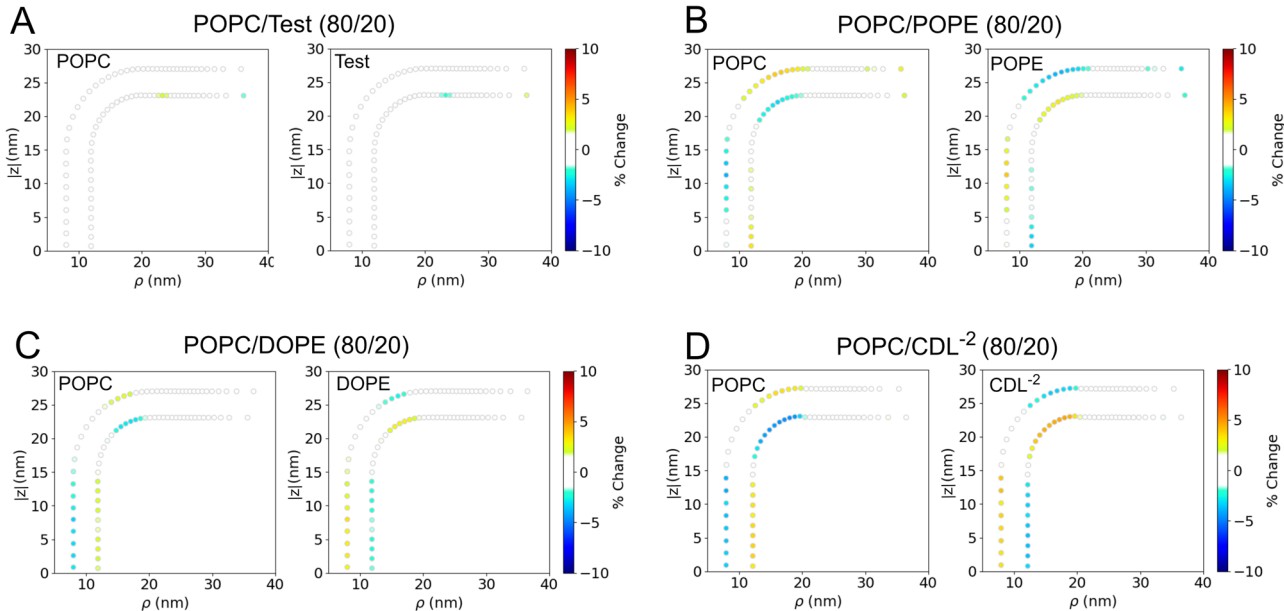

**Fig. 4 Averaged lipid partitioning analysis of the studied binary IMM model systems during final 1 μs. A** POPC, (**B**) POPC/POPE, (**C**) POPC/DOPE and (**D**) POPC/CDL$^{-2}$. In all figures, any percentage change below 1.5% was set to zero.

DOPE. In comparing CDL$^{-2}$ vs. PE lipids in the binary systems, CDL$^{-2}$ displays a similar level of accumulation in the inner leaflet cylinder (4.1%) as POPE but has considerably greater accumulation in the outer leaflet junction (4.7%) than either DOPE or POPE. This finding may indicate a sensitivity toward Guassian curvature features and not only mean curvature.

Given that POPE/DOPE and CDL lipids both have preferences for negative curvature regions, we further explored this concept in ternary component bilayer systems that contained both PE and CDL lipids in a POPC background (Fig. 5). We find that the general observations from the binary systems are reinforced in the ternary systems. In the POPC/POPE/CDL$^{-2}$ system (Fig. 5A), CDL has more significant accumulation, over a greater area range, in both the outer junction and inner cylinder than POPE. Similiar findings are observed in the POPC/DOPE/CDL$^{-2}$ system (Fig. 5B). Lastly, when the charge on the CDL headgroup is reduced to -1, in a POPC/POPE/CDL$^{-1}$ system (Fig. 5C), the partitioning character of CDL is greatly enhanced compared to POPE. In terms of the maximum localized accumulation we see that CDL has the most significant accumulation in all the ternary systems, and has higher accumulations than in the binary systems: POPC/POPE/CDL$^{-2}$: CDL$^{-2}$ has 5.3% accumulation vs. 2.8% for POPE; POPC/DOPE/CDL$^{-2}$: CDL$^{-2}$ has 4.9% accumulation vs. 2.1% for DOPE; POPC/POPE/CDL$^{-1}$: CDL$^{-1}$ has 6.7% accumulation vs. 4.1% for POPE.

In addition to examining the time-averaged lipid localization properties, MD allows us to examine the time-dependent behavior of these systems. We can visualize the lipid dynamics using the same coordinate mapping approach (Fig. S7), to allow us to observe with a high degree of spatial resolution where different lipids are (re)locating. These analyses are presented in Figs. S9–S17. While one can qualitatively discern the partitioning behavior, the high spatial and temporal resolution in these analyses makes for difficult quantitative evaluation. To provide a more clear depiction of the lipid dynamics, we performed an analysis where we computed the lipid species concentration changes over the entire regions (i.e., compartments) of the IMM through the MD simulations. This compartmental analysis is illustrated for the POPC/POPE, POPC/CDL$^{-2}$, and the POPC/

POPE/CDL$^{-2}$ system in Figs. 6, 7, 8, respectively, and the additional systems are presented in Figs. S18–S23.

From the compartmental analysis, we can compute a quantitive enrichment/depletion factor ($F$) for each region for both leaflets, which in essence, is just the percentage change in the amount of a given lipid type in a given region. $F$ is defined by eq. (1) and eq. (2)

$$F = (\langle LF_{m,ta \to tb} \rangle - LF_{m,t=0}) \cdot 100, \quad (1)$$

$$LF_i = \frac{n_i}{\sum_{i=1}^{i=2 \text{ or } 3} n_i}, \quad (2)$$

where $LF_m$ is the minor lipid component fraction, and $n_i$ is the number of lipids of type $i$ present in the particular compartment of the IMM model. The sum in the denominator of eq. (2) represents the total number of lipids in a particular compartment, and therefore, the sum is over two components for binary systems and three for ternary systems. The reported results are based on the final 1 μs simulation data, i.e., 3 μs to 4 μs ($ta \to tb$). The errors were estimated using block averaging with a block size of 80 ns, and the results are presented in Table S3.

In the case of the control system (POPC/Test), we observe reasonably good convergence properties by having a minimal (< 1%) amount of statistically significant enrichment in any region (Fig. S18, Table S3). For the binary systems (Figs. 6, 7, and S19), we see that the minor component lipids have accumulation in the inner cylinder and outer junction regions. Based on the enrichment factors, CDL$^{-2}$ has the greatest enrichment in the inner cylinder (3.3%), while DOPE and POPE have lower enrichment in this region (2.5%). Likewise in the outer junction, CDL$^{-2}$ has the greatest accumulation (3%), followed by DOPE (2.9%), and POPE has the least accumulation (2.7%).

In the ternary systems (Figs. 8, S20–S21), we see that CDL$^{-2}$ has stronger partitioning to negative curvature than POPE or DOPE (Fig. 8) as evidenced by inner cylinder accumulation, outer junction accumulation, and outer cylinder depletion. This is an interesting result, which implies that the complexity of the ternary system and the interplay of the various lipids cannot be directly predicted from the binary system results. The CDL

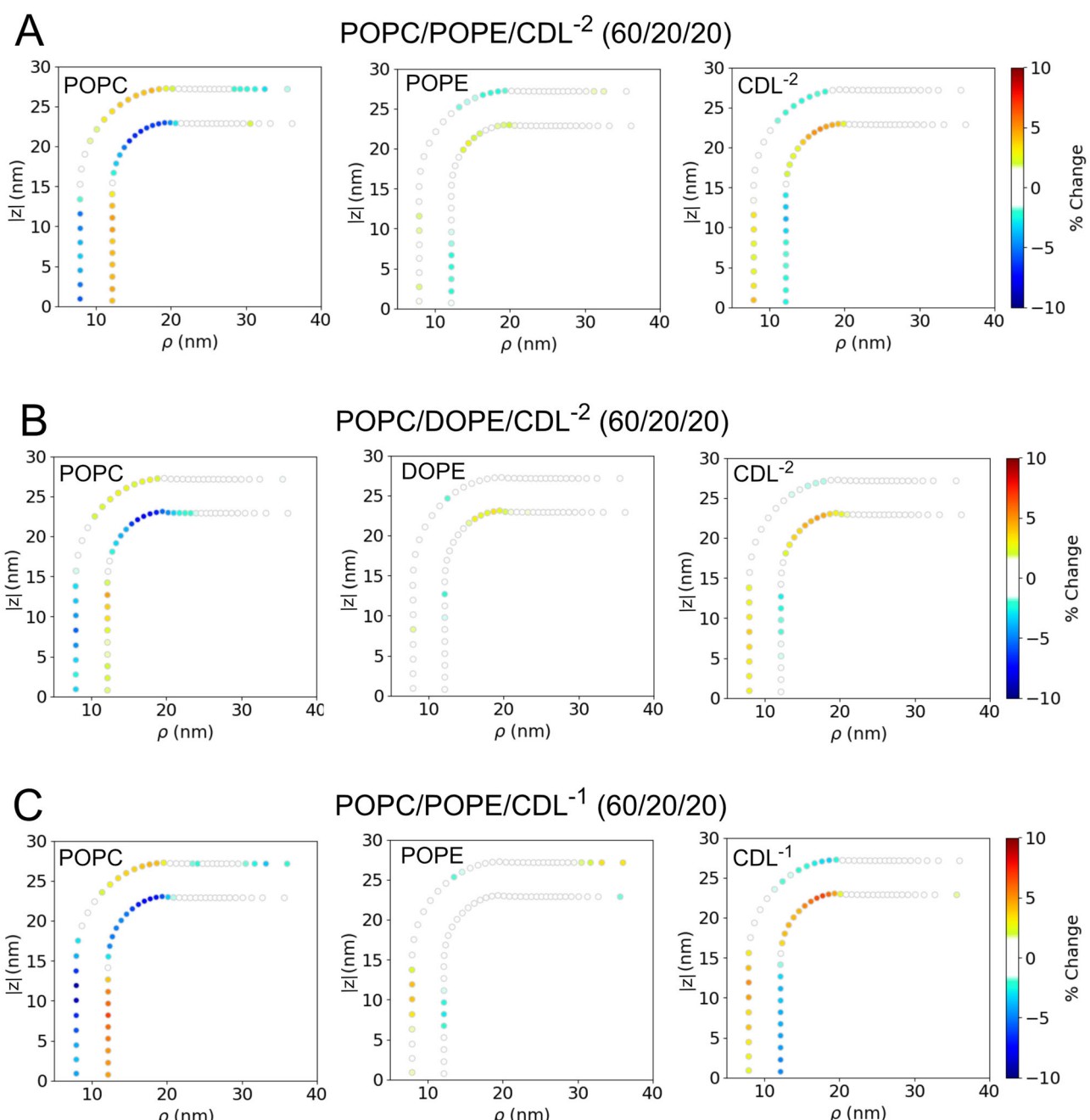

**Fig. 5 Averaged lipid partitioning analysis of the studied ternary IMM model systems during final 1 µs. A** POPC/POPE/CDL$^{-2}$, (**B**) POPC/DOPE/CDL$^{-2}$, and (**C**) POPC/POPE/CDL$^{-1}$. In all figures, any percentage change below 1.5% was set to zero.

partitioning effects are considerably stronger when the headgroup charge is reduced to -1 (Fig. S21), where maximal accumulation (across all $r_{cyl} = 10$ nm systems) of 4.3% is observed in the outer junction. These results related to CDL and POPE are in agreement with the previously reported observations in buckled membranes[22,24]. The partioning propensity of CDL$^{-2}$ in the presense of POPE vs. DOPE shows that the partioning effect is greater in the presence of POPE. In the POPC/POPE/ CDL$^{-2}$ system, CDL$^{-2}$ has significantly greater accumulation than POPE in the outer junction (2.7% vs. 1.5%) and inner cylinder (3.0% vs. 1.1%), and greater depletion in the outer cyliner (-3.0% vs. -1.8%). The same comparisions in the POPC/DOPE/ CDL$^{-2}$ system display very marginal erichment/depletion of CDL over DOPE (outer junction: 2.1% vs. 2.0%; inner cylinder: 2.5% vs. 0.7%; outer cylinder:-0.9% vs. -1.1%).

As a final consideration, we constructed two additional systems to evaluate the effect of altering the degree of curvature. We consider the POPC/POPE/CDL$^{-2}$ to be the biologically most relevant system (see Discussion) and therefore we constructed systems in which the cylinder radius was increased to 15 nm and decreased to 5 nm. We computed the partitioning behavior in these additional systems, the time course dynamics are displayed in Figs. S16, S17 and the compartmental partitioning profiles are presented in Figs. S22, S23. We compare the partitioning in the inner cylinder and outer junction regions in Fig. 9 for systems with different cylinder radii, which are the compartments where minor component lipids have the most significant accumulation. The effect of curvature is most noticeable in the inner cylinder where CDL$^{-2}$ displays a greater response to increasing curvature (smaller radius) than POPE.

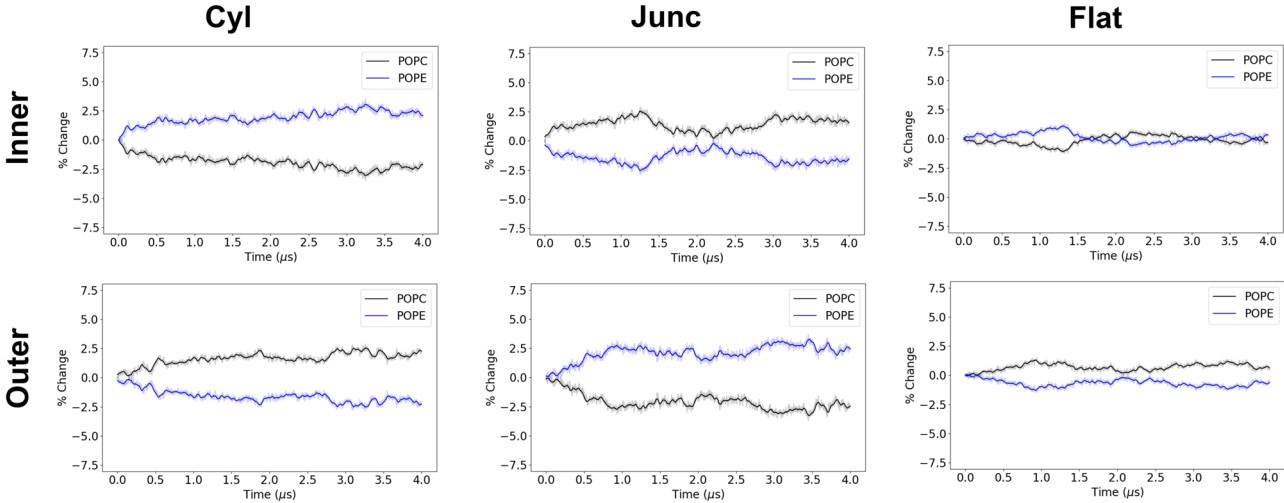

**Fig. 6 Compartmental analysis for POPC/POPE (4:1) IMM system.** The percentage change for all lipids in all compartments of both inner and outer leaflets is calculated over the 4 $\mu$s trajectory. The solid lines are running averages computed using a 40 ns window, and the transparent line is the raw data.

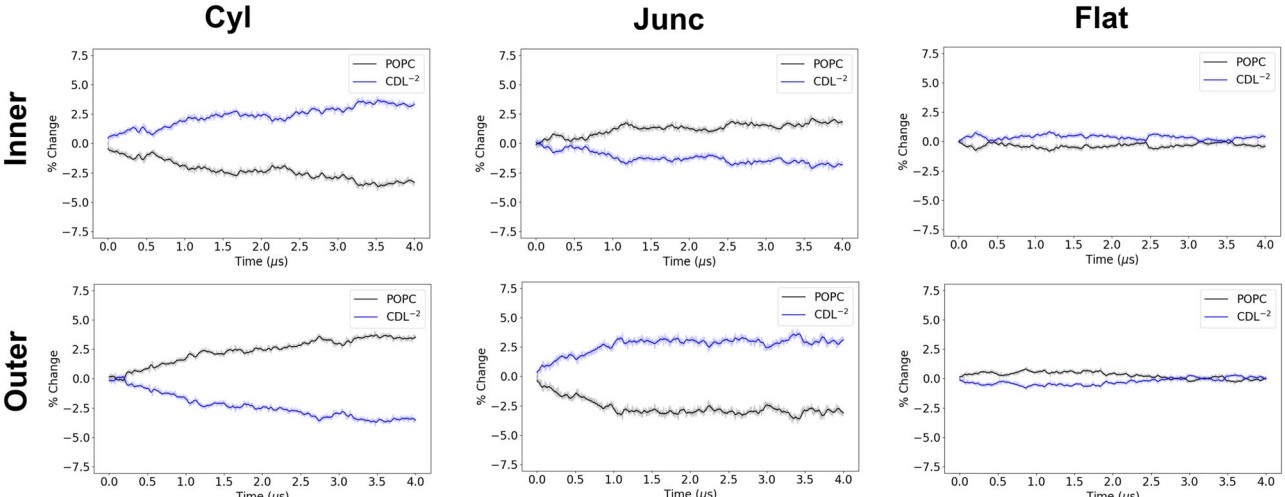

**Fig. 7 Compartmental analysis for POPC/CDL$^{-2}$ (4:1) IMM system.** The percentage change for all lipids in all compartments of both inner and outer leaflets is calculated over the 4 $\mu$s trajectory. The solid lines are running averages computed using a 40 ns window, and the transparent line is the raw data.

To our knowledge, there has not been a substantial investigation into whether Gaussian curvature affects lipid localization properties. Our study provides an opportunity to examine this question and try to discern if Gaussian curvature is a significant driving force for lipid localization or if mean curvature is the dominant property that lipids sense. To this end, we performed linear correlation analysis to determine the Pearson correlation coefficient ($r_P$) between the enrichment factors and either mean or Gaussian curvature. For each system there are six compartments where both $H$ and $K$ have been measured (Table S2), which can then be correlated with the enrichment factors (Table S3). We observe in Fig. 10 that the enrichment factors have a strong negative correlation with the mean curvature, which is logical since the minor component lipids are all accumulating in regions of negative mean curvature. Across all system the average $r_P^{mean}$ is -0.81, with maximal correlation observed for CDL$^{-2}$ in the binary system (-0.93) and for POPE in the ternary system containing CDL$^{-1}$ (-0.94). We also observe there is a consistent (but weaker) negative correlation between the enrichment factors and the Gaussian curvature. All systems display a negative correlation, with an average value of $r_P^{Gauss} = -0.32$ and maximal correlation

is for POPE in the ternary system with a 5 nm cylinder radius (-0.58).

## Discussion

In the current study, we quantitatively evaluated the forces on dummy particles and the partitioning of lipids in a curved bilayer system representative of an IMM cristae. We observed that the measured forces on the dummy particles group matched the applied constant forces (see Fig. S6), validating that the dummy particles can act as force sensors. The membrane properties were found to be minimally perturbed up to a max force of 50 kJ mol$^{-1}$ nm$^{-1}$ (Figs. S3–S5). In the IMM systems, the measured average forces on the dummy particles did not exceed 10 kJ mol$^{-1}$ nm$^{-1}$ (Fig. 2). Therefore we expect the lipid dynamics to be near native during these simulations.

The enrichment of PE and CDL lipids in the regions of the outer junction and the inner cylinder region of the IMM model was expected due to their intrinsic negative curvature preference and the negative mean curvature of these IMM regions. The localization of these lipids is presumably driven by the minimization of curvature frustration energy. The accumulation or

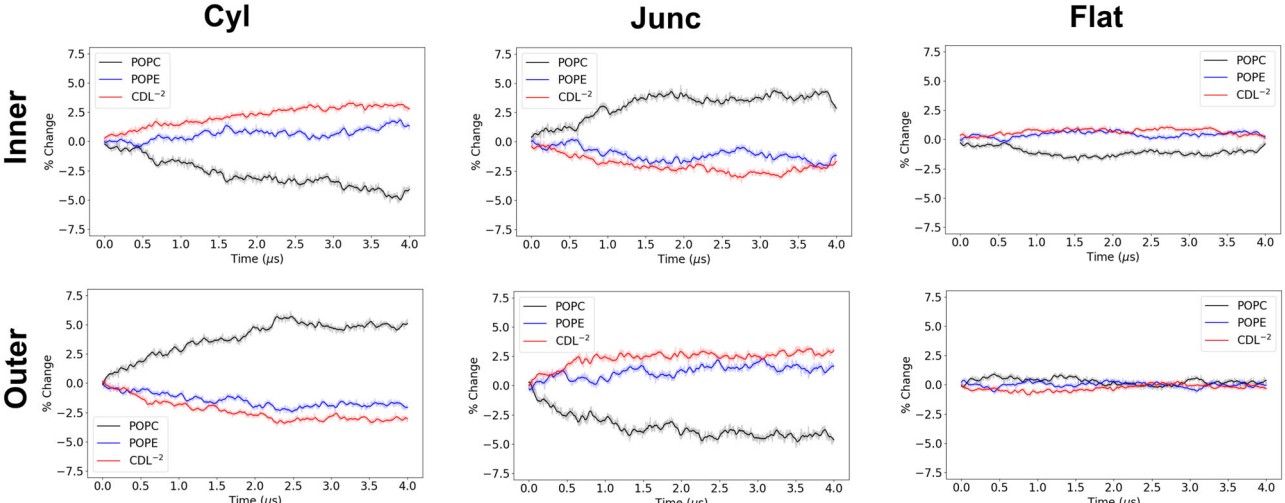

**Fig. 8 Compartmental analysis for POPC/POPE/CDL$^{-2}$ (3:1:1) IMM system.** The percentage change for all lipids in all compartments of both inner and outer leaflets is calculated over the 4 $\mu$s trajectory. The solid lines are running averages computed using a 40 ns window, and the transparent line is the raw data.

depletion of PE and CDL lipids is counterbalanced by the redistribution of PC lipids. There is an entropic cost to demixing, which is compensated for by a combination of reducing bending energetics and also enthalpically favorable interactions. While experimental studies have indicated favorable lipid-lipid clustering is required to overcome the entropic loss of demixing[27,28], simulations studies have not observed this phenomenon in CDL containing systems. In previous studies using the MARTINI force-field in buckled membranes, CDL enrichment was observed without CDL clustering[22,24]. While CG models may lack key features needed to predict specific lipid-lipid interactions (e.g., hydrogen-bonding), atomistic simulations of CDL containing mitochondrial model membranes also did not predict CDL clustering[21]. Based on electrostatic considerations, it is challenging to rationalize how di-anionic CDL would form direct and stable CDL-CDL interactions. However, it is possible that more complex and heterogenous interactions (e.g., CDL-PE-CDL) are providing favorable enthalpy to bias towards segregated states. Given that CDL clustering has not been observed in previous studies with the same model, we believe curvature energetics are the dominant force in driving lipid enrichment/depletion in the current study.

We have designed a system which has curvature features comparable to biological structures, where previous simulations studies on lipid sorting have used buckled membranes[20,22,24,38] and membrane tethers[23], which both have higher degrees of curvature and geometries less representative of naturally occurring, stable biological structures. In those previous studies, a greater degree of CDL enrichment was observed in negative curvature regions compared to the current study, but this is likely reflective of the greater degree of curvature in those previous studies. Supporting this notion, we considered systems with decreased (and increased) cylindrical radii and found that tighter curvatures drive more significant partitioning (Figs. 9, S22). We observe relatively slight but statistically significant accumulations, at most up to 5% change in total lipid composition, which represents a 25% increase in CDL concentration (from 20% to 25% in CDL$^{-1}$ system). The range of systems explored allowed us to make inferences about head group type, charge, and tail composition. Our results indicate the following rankings in terms of negative curvature preference: CDL$^{-1}$ > CDL$^{-2}$ > DOPE > POPE > POPC. CDL$^{-1}$ has a smaller effective headgroup than CDL$^{-2}$ giving it a more negative spontaneous curvature.

The relative rankings of DOPE, POPE, and POPC are consistent with measured monolayer spontaneous curvature values[39]. The simulation results support that increasing acyl chain tail saturation increases accumulation in negative curvature regions, presumably by increasing the occupied tail volume. While the result of CDL being more prone to negative curvature accumulation than DOPE is consistent with previous results, the finding does appear at odds with the measured spontaneous curvatures[36]. However, when examining the binary systems, DOPE does accumulate to an equivalent extent in the outer junction as CDL$^{-2}$, and has greater depletion in the inner junction and outer cylinder. Nevertheless, in the ternary POPC/DOPE/CDL$^{-2}$ system, CDL outcompetes DOPE and has greater accumulation in negative curvature, implying a complex interplay between the multiple lipid species. Based on lipid tail compositional analysis, 16:0 chains are more common in mitochondria than 18:1 chains[40] indicating POPE should be present at higher concentrations than DOPE in the IMM. The point is that the more biologically relevant comparison may be POPE vs. CDL, where we clearly see that CDL has greater negative curvature sensing.

While it has been shown that proteins and peptides can sense the curvature of membranes[41,42], including Gaussian curvture[43], this study aimed to evalute the mean and Gaussian curvature sensitivity of the different lipid species, which to our knowledge has not been previously evaluated. Based on linear correlation analysis, we find that the lipids are sensing and responding to the mean curvature but also have a consistent but weaker correlation with Gaussian curvature properties, Fig. 10. It has been recognized that modeling cellular membranes in greater complexity, including more chemical components, bilayer asymmetry and ranges of curvature are challenges that must be addressed to gain greater insights into membrane-associated phenomena[44–46]. This study extends our knowledge of lipid curvature sensing and sorting to a mitochondra innner membrane model that has realistic curvature features. Our results support that CDL can outcompete PE lipids for accumulation in regions of mean negative curvature, further indicating how the unique properties CDL are able to contribute to mitochondrial structure and function.

## Methods

**System setup and simulation details.** Simulations presented in this work were conducted using the GROMACS 2020 package[47]

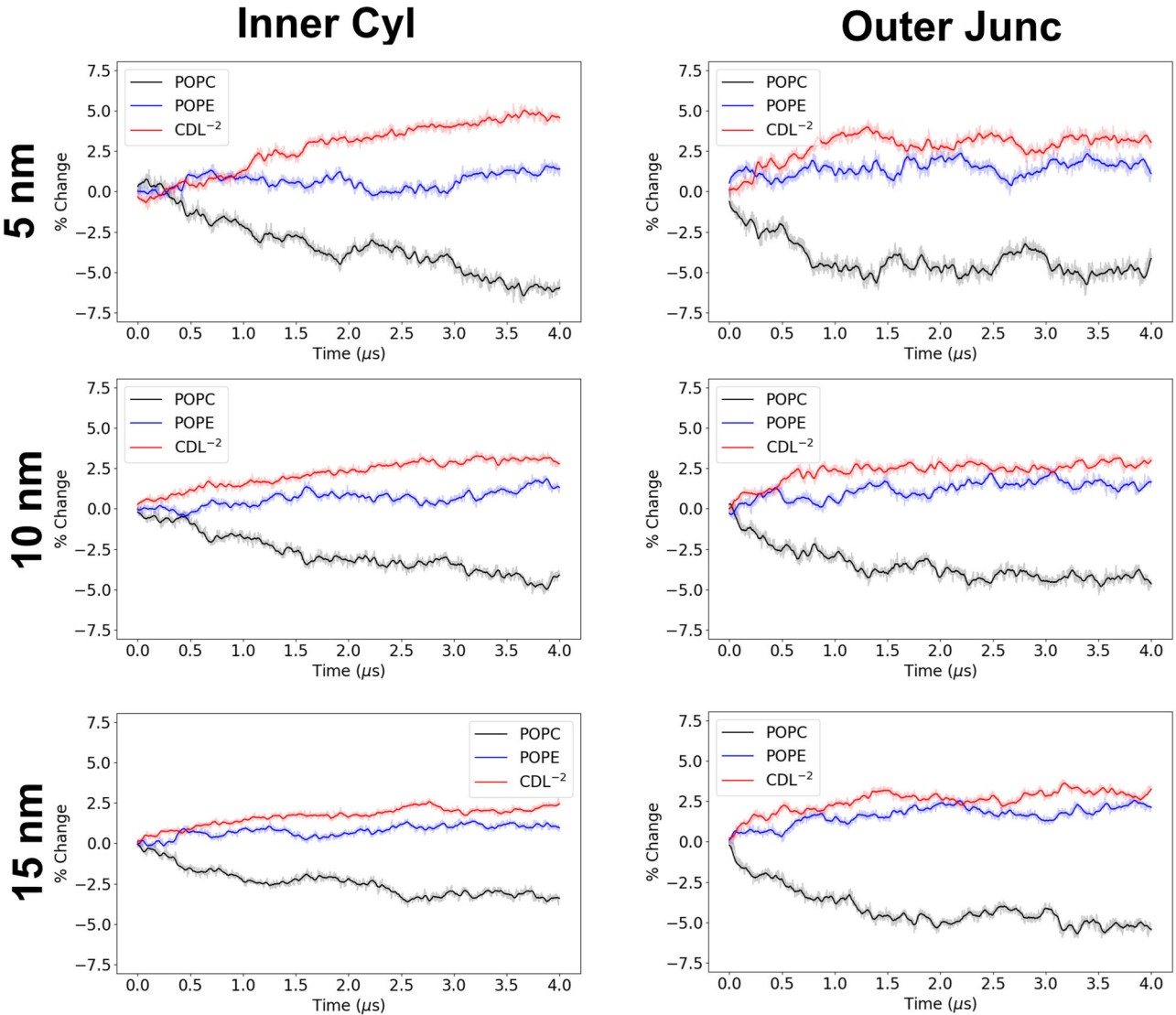

**Fig. 9 Compartmental analysis for POPC/POPE/CDL$^{-2}$ (3:1:1) IMM system at different cylinder radii.** The inner cylinder (left) and outer junction (right) regions are presented for systems with cylinder radii ranging from 5 nm to 15 nm. The percentage change for all lipids in all compartments of both inner and outer leaflets is calculated over the 4 $\mu$s trajectory. The solid lines are running averages computed using a 40 ns window, and the transparent line is the raw data.

using the MARTINI 2.2 coarse-grained (CG) force field with the standard (non-polarizable) water model[48]. The standard model for CDL lipids in MARTINI uses five beads to model each acyl chain. However, we have found that modeling the acyl chains of CDL using four beads provides a model improvement for accurately predicting bilayer properties. This 4-bead model is validated in Fig. S24, where the bilayer thickness and lateral pressure profiles are shown to better match all-atom data, compared to the 5-bead model. Hence, in all simulations presented here, the 4-bead model is used. The parameters for this model are provided in the Supplementary Data 1 (CDL$^{-2}$) and Supplementary Data 2 (CDL$^{-1}$) files. Further, it should be noted that all other lipids used in this study are modeled with 4-bead acyl chains in MARTINI.

**IMM simulation details**. Flat bilayer systems were generated using the insane.py script[49] and subjected to two-step minimization followed by four short NVT runs of 100,000 steps each with successively increasing time steps from 2 fs to 20 fs (i.e., 2, 5,

10, 20). Subsequently, systems were pre-equilibrated for 2 ns under NPT conditions using a Berendsen barostat[50] and v-rescale thermostat[51]. A final equilibrium simulation of 100 ns was performed under NPT conditions, using semi-isotropic pressure coupling with a coupling constant of 12 ps and the pressure set to 1 bar using the Parrinello-Rahman barostat[52]. The equilibrated bilayer structure was used as an input for the BUMPy tool. The following BUMPy input line was used to generate all $r_{cyl} = 10$ nm IMM systems (input values are in Å): *bumpy.py -f flatinput.gro -z 10 -s double_bilayer_cylinder -o bumpy-sys.pdb -p topol.top -n index.ndx -g l_cylinder:300 r_cylinder:100 r_junction:100 l_flat:560 –gen_dummy_particles –dummy_grid_thickness 50*. The list of simulation systems and corresponding lipid compositions are listed in Table S1. System A was a control system in which 20% of the POPC lipids were renamed "Test" without changing any of the interaction parameters. To maintain the IMM shape, the bilayers were bracketed on both sides with a layer of dummy particles (see Fig. 1C). The thickness between the layers of the dummy particles was 50 Å, and the lateral spacing between particles was 5 Å. The dummy particles interact with a repulsive

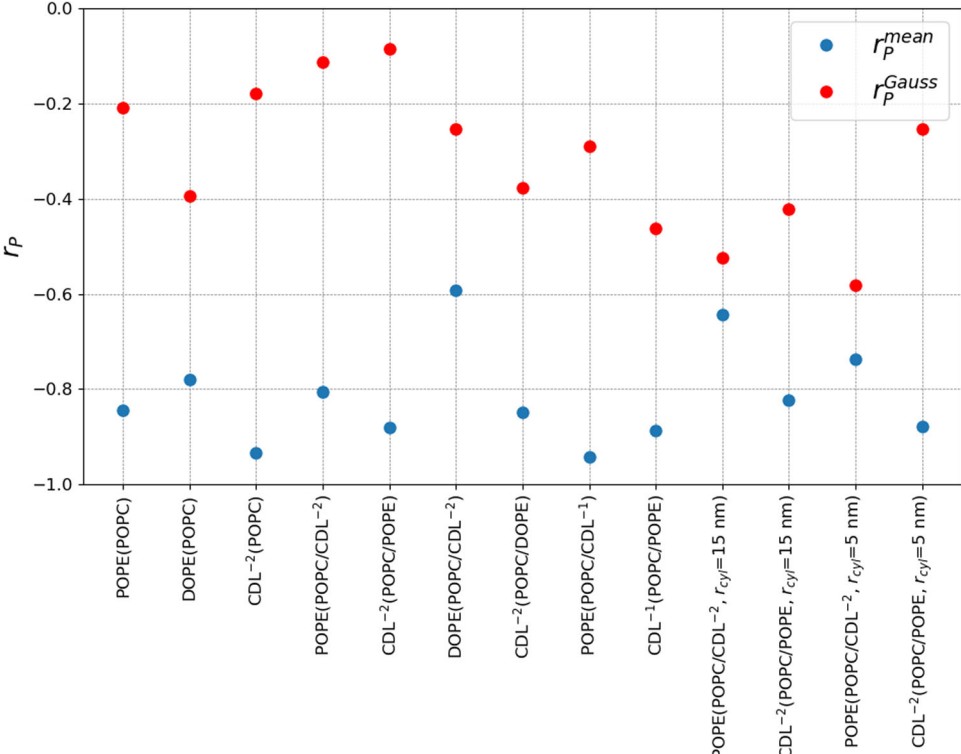

**Fig. 10 Correlation analysis.** The Pearson correlation coefficient between the enrichment factors and the mean (blue) and Gaussian (red) curvatures are shown for all minor component lipids. On the x-axis, each system is labeled with the minor component and the other components in parenthesis. All systems have a 10 nm cylinder radius ($r_{cyl}$), unless otherwise noted.

force on lipid tail beads through a Lennard-Jones potential, with the repulsive $C_{12}$ parameter set to 0.0258 kJ nm$^{12}$ mol$^{-1}$ and the attractive $C_6$ parameter set to 0[32]. The *gmx solvate* tool of GROMACS was used to add water molecules and the systems were charge neutralized by adding MARTINI sodium ions using the *gmx genion* tool. The electrostatic interactions were modeled with the reaction field method with a relative dielectric constant of 15 and a reaction field dielectric constant of infinity for the long and short-range regimes, respectively. A cut-off distance of 11 Å was used for the short-range electrostatics and non-bonded interactions, and the potential-shift-Verlet scheme was used to cut off the non-bonded forces following the protocol of de Jong et al.[53].

All the IMM systems, shown in Table S1, were subjected to a two-step energy minimization followed by a four-step equilibrium protocol under NVT ensemble conditions. In brief, during the first step of energy minimization, the steepest descent algorithm and the soft-core potentials for short-ranged interactions were employed to allow some tolerance for overlapping atoms. In the second step, energy minimization was performed using the steepest descent algorithm without the soft-core potentials. Following energy minimization, five short equilibrium (100,000 steps) simulations were performed using the MD integrator while successively incrementing the time step from 2 fs to 20 fs, while releasing the restrains on the membrane lipids from 50 to 0 kJ mol$^{-1}$ nm$^{-2}$. The dummy particles were fixed by freezing the dummy particles group in the *x*, *y*, and *z* dimensions. Four microsecond-long NVT production MD simulations were performed with a time step of 20 fs. The temperature of the systems were maintained at 310 K using the v-rescale thermostat. The obtained trajectories were used to analyze the lipid partitioning and curvature properties using in-house scripts and the MemSurfer Python tool[37].

**Evaluation of forces on dummy particles**. To evaluate forces on dummy particles, constant force simulations were performed using the GROMACS integrated pull-code method. The approach was validated on a flat bilayer system containing 200 CG lipids (POPC/DOPE (4:1)), generated using insane.py[49], and a grid of dummy particles was placed above the upper leaflet using the BUMPy tool[32] with a lateral spacing of 5 Å (see Fig. S2). The generated system was subjected to a two-step energy minimization followed by a four-step equilibrium protocol under the NVT condition, as described above. The system was then relaxed for 50 ns under NPT ensemble condition using the Berendsen barostat[50] and v-rescale thermostat[51]. The dummy particles were restrained with a harmonic force constant of 1000 kJ mol$^{-1}$ nm$^{-2}$. After the completion of equilibrium simulations, a series of constant force pulling simulations were performed by applying the following different force constants in independent simulations to the center of mass (COM) of the bilayer for a period of 250 ns: 0, 1, 5, 10, 20, 50, 100, 250, 500, and 1000 kJ mol$^{-1}$ nm$^{-1}$. The direction of the force was in the positive *z*-direction such that the COM of the bilayer was pulled into the layer of dummy particles. The effect of pressure coupling (NVT vs. NPT) and the whether the dummy particles were restrained or frozen were examined. During NPT simulations, when dummy particles were harmonically restrained, reference coordinate scaling of all reference positions was performed. The final 100 ns of the trajectories were used to calculate bilayer properties and estimate the forces acting on the dummy particles. In both NVT and NPT simulations, the v-rescale thermostat was used, and in NPT simulations, the Parrinello-Rahman barostat[52] was used. The temperature of the systems were maintained at 310 K with a coupling constant $\tau_t = 1$ ps. Semi-isotropic pressure coupling in the NPT systems was used to maintain 1 bar pressure in the membrane normal and lateral directions with a time interval of $\tau_p = 12$ ps and a compressibility

of $3 \times 10^{-4}$ bar $^{-1}$. Additionally, lateral pressure profiles were calculated by computing the 3D stress tensor ($\sigma$) using the GROMACS-LS package[54,55]. The stress tensor is averaged in the $x$ and $y$ dimensions and the lateral pressure profile is computed as $\pi(z) = P_L(z) - P_N(z)$, where $P_L = -(\sigma_{xx} + \sigma_{yy})$ and $P_N = -\sigma_{zz}$. Profiles were computed using the final 1 μs of simulation data; $\pi$ and $P_N$ are plotted for all IMM systems with $r_{cyl} = 10$ nm in Fig. S8.

**Reporting summary.** Further information on research design is available in the Nature Portfolio Reporting Summary linked to this article.

## Data availability

Simulation input files, example data and analysis codes can be accessed from: https://github.com/MayLab-UConn/IMMLipidDynamics. Additional data will be available upon request.

## Code availability

The modeled inner mitochondrial membrane systems were generated by using BUMPY tool (https://github.com/MayLab-UConn/BUMPy). Molecular dynamics (MD) simulation data was generated using GROMACS version 2020.4 (https://manual.gromacs.org/documentation/2020.4/download.html) with MARTINI 2.2 force field parameters (http://cgmartini.nl/index.php/force-field-parameters/lipids). Data analysis was performed by using GROMACS in-built analysis tool, in-house developed python scripts, and MemSurfer tool for the curvature property analysis (https://github.com/LLNL/MemSurfer). Custom codes can be accessed at the May Lab github repository (https://github.com/MayLab-UConn/IMMLipidDynamics).

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

## Acknowledgements

This work was supported by the National Institutes of Health through grant number R35-GM119762 (to E.R.M.). Computational resources for this work have been provided through the University of Connecticut Storrs HPC center. The Supporting Information contains Tables S1–S3 and Figs. S1–S24. The MARTINI GROMACS parameter (.itp) are provided in Supplementary Data 1 and Supplementary Data 2 files for the CDL$^{-2}$ and CDL$^{-1}$ models using 4 beads per acyl chain, respectively.

## Author contributions

V.K.G. conducted simulations, analyzed data and wrote the manuscript. K.J.B. designed the study and developed analysis methods. E.R.M. designed the study, analyzed data and wrote the manuscript.

## Competing interests

The authors declare no competing interests.
