## [Peer Review File · Communications Biology]

Reviewers' comments:

Reviewer #1 (Remarks to the Author):

The manuscript "Curvature sensing lipid dynamics in a mitochondrial inner membrane model" by Golla et. al. is a rigorous attempt to demonstrate the curvature dependent sorting of various lipid molecules using martini based framework. In particular, the authors captured strong preference of cardiolipin towards negatively curved region of the membrane which is consistent with previous studies. With a biologically relevant cristae model, the authors established that the lipid partitioning is mainly dominated by mean curvature while there exist a weaker correlation with gaussian curvature. Overall, the approach is reasonable, and the conclusions are supported by the data. The findings of this paper will be useful to the overall biophysics' community. However, I have few concerns that are listed below.

1. From Fig-1A, it seems like the membrane is non periodic or discontinuous along Z axis due to solvent padding while continuous along X and Y axis. I recommend putting an outline to depict the overall simulation box (like 'pbc box' command in VMD) boundary to understand the periodicity of the system better. I was also wondering whether the authors can make the system continuous along Z dimension by considering one layer (upper) among the flat bilayer on top and other layer (lower) at bottom and adjust the z dimension of the box just to fit the Z length of the membrane. In this way the membrane remains continuous along Z dimension, total number of CG particle will be reduced, and the usage of the dummy atoms will be minimal.
2. I assume solvent molecules flow between inside and outside of the cylinder. Since dummy atoms are involved to maintain the overall shape of the membrane, can authors verify the stability of the system by computing the radial variation of the local stress profile from inside of the cylinder to the outside of the cylinder like Fig-1 of Ollila et. al. [PRL 102, 078101 (2009)]?
3. It would be interesting to see the curvature dependent sorting when radius of curvature is varied. Will cardiolipin's propensity to accumulate at the negatively curved region be amplified if a similar membrane model with smaller radius of the cylindrical region (enhanced curvature) is considered?
4. The authors have so far considered total 6 kind of curvature regions of the membrane. The authors have not considered a situation when mean and gaussian curvature both are significantly greater or less than zero like a situation in spherical vesicle. Can the authors model a spherical vesicle on one end of the cylinder instead of a flat bilayer while keeping the other end remain unperturbed? It would be interesting to see curvature dependent sorting in this kind of set up.
5. The 2D polar coordinate representation seems confusing to me. Why did the authors consider only one (top) flat end and junction region? Shouldn't there be another flat and junction region near $z = 0$? Did the authors consider middle region of the cylinder as $z = 0$? In that case Y axis should be labelled as $|z|$ instead of z.

Reviewer #2 (Remarks to the Author):

This study investigates curvature-based lipid sorting in the inner mitochondrial membrane (IMM) using CG Martini lipid force field. The authors used their previously developed approach (JCTC 2018) to build a mitochondrial membrane, in which bilayers are bracketed on both sides by a layer of dummy atoms, which seem to serve as a flat bottom harmonic restraint to prevent the bilayer deviates from the desired geometry. They then tested the impact of dummy atoms on the curved system by evaluating the forces on dummy atoms to ensure the least perturbation on the lipid dynamics.

The lipids simulated were POPC, POPE, DOPE, and cardiolipin (CDL). They found POPE/DOPE and CDL localize in the higher negative curvature regions, consistent with the intrinsic negative curvature of PE and CDL. Correlation analysis shows the lipids are more sensitive to the mean curvature, but also, to a less extent, Gaussian curvature.

Overall, this simulation system is at a biologically relevant length scale and the analysis is carefully done. It is also quite interesting to see that the charge on CDL has minimal effect on lipid sorting as the curvature energetics are the dominant force in the current study. The main limitation of the current study is that only a single IMM shape and size were simulated. It is likely that some variation exists in biology. Comparison across several systems, for example, with variation in the radius of the cylinder or curvature of the junction will significantly improve the current manuscript. Other factors such as changes in lipid rigidity may also affect curvature sorting, which could be potentially investigated here and make the conclusion more comprehensive.

We are thankful for the efforts of the reviewers and editors to improve our manuscript and appreciate the general enthusiasm and acknowledgment of the quality of our work. We have responded point-by-point to the reviewers comments below and have highlighted changes to the manuscript text by using red fonts.

Reviewer #1 (Remarks to the Author):

The manuscript “Curvature sensing lipid dynamics in a mitochondrial inner membrane model” by Golla et. al. is a rigorous attempt to demonstrate the curvature dependent sorting of various lipid molecules using martini based framework. In particular, the authors captured strong preference of cardiolipin towards negatively curved region of the membrane which is consistent with previous studies. With a biologically relevant cristae model, the authors established that the lipid partitioning is mainly dominated by mean curvature while there exist a weaker correlation with gaussian curvature. Overall, the approach is reasonable, and the conclusions are supported by the data. The findings of this paper will be useful to the overall biophysics’ community. However, I have few concerns that are listed below.

Thank you very much for your time in reviewing the manuscript and providing us with your constructive comments and suggestions.

1. From Fig-1A, it seems like the membrane is non periodic or discontinuous along Z axis due to solvent padding while continuous along X and Y axis. I recommend putting an outline to depict the overall simulation box (like ‘pbc box’ command in VMD) boundary to understand the periodicity of the system better. I was also wondering whether the authors can make the system continuous along Z dimension by considering one layer (upper) among the flat bilayer on top and other layer (lower) at bottom and adjust the z dimension of the box just to fit the Z length of the membrane. In this way the membrane remains continuous along Z dimension, total number of CG particle will be reduced, and the usage of the dummy atoms will be minimal.

Response: A new Fig-1A has been made to show the periodic boundary box. The reviewer is correct that there is solvent padding above/below the flat regions, which are continuous with the periodic images in X and Y directions. As far as the suggestion about making the box continuous in Z, either we do not believe it is feasible or we are not understanding the suggestion. I believe the suggestion is to remove the lower flat region, have the cylinder extended through to the end of the box and have the upper flat region be at the top of the box. This scenario would not be feasible because the cylinder would be “entering” the flat regions from the top and simultaneously “exiting” the flat region from below. There would be a mismatch of leaflets, there would be 4 cylinder leaflets (2 each from the entering and exiting cylinders) but only two flat region leaflets. Therefore you could not have continuity between both cylinders and the flat regions in this scenario.

2. I assume solvent molecules flow between inside and outside of the cylinder. Since dummy atoms are involved to maintain the overall shape of the membrane, can authors verify the stability of the system by computing the radial variation of the local stress profile from inside of the cylinder to the outside of the cylinder like Fig-1 of Ollila et. al. [PRL 102, 078101 (2009)]?

Response: Thank you for your suggestion. However there is not flow of solvent between the inside and outside of the cylinder. These are separate “compartments” and in analogy to the mitochondria the region inside the cylinder would correspond to the intermembrane space while the region outside the cylinder would correspond to the matrix. We should have made this more clear and we

have added a few sentences to the second to last paragraph of the introduction to highlight this point. We have added:

“Another important feature of our IMM model is that it segregates the system into two separate aqueous compartments, analogous to the mitochondrial matrix and IMS. We envision the region on the inside of the cylinder as the IMS and the region outside the cylinder as the matrix. While we do not exploit this feature in the current study, in future work we could examine the effect transmembrane potentials by varying the ionic species in the different solvent compartments.”

To check the stability of the systems the lateral pressure profiles (LPP) have been estimated for the IMM systems and shown in supporting information. To be noted, the use of NVT ensemble conditions and the unusual geometry of the system make the interpretation of the pressure profiles challenging. We do observe the z-component of pressure is relatively constant and the LPPs show near zero values in the cylindrical regions (Z=20-40 nm). LPPs and P_N for all IMM systems are now shown in Figure S7.

3. It would be interesting to see the curvature dependent sorting when radius of curvature is varied. Will cardiolipin's propensity to accumulate at the negatively curved region be amplified if a similar membrane model with smaller radius of the cylindrical region (enhanced curvature) is considered?

Response: We agree and we do believe that cardiolipin (CDL) would have stronger accumulation in negatively curved regions if smaller radii of curvature segments were simulated. Indeed we have seen this in buckled membranes (Boyd, Alder, May Langmuir 2017), where the curvatures are much more extreme than in the current study and we see enrichment of CDL in the most tightly curved regions of up to 100% (going from 20 mol% bulk to 40 mol% in these regions). In the context of the current study, it would be a significant undertaking to vary the geometry and by going to smaller cylinder radii it would move the simulation system out of the regime of “biologically realistic”. However, from a physical perspective it certainly is a valuable study and something we will consider in follow-up work.

4. The authors have so far considered total 6 kind of curvature regions of the membrane. The authors have not considered a situation when mean and gaussian curvature both are significantly greater or less than zero like a situation in spherical vesicle. Can the authors model a spherical vesicle on one end of the cylinder instead of a flat bilayer while keeping the other end remain unperturbed? It would be interesting to see curvature dependent sorting in this kind of set up.

Response: If I understand correctly the suggestion is to model one end of the cylinder with a hemisphere, which would represent a cristae tip and also model a region with non-zero gaussian and mean curvature. The current study does include a region which has significant non-zero gaussian and mean curvature which is the junction region. The outer leaflet has mean curvature comparable (in magnitude) to the outer cylinder and has negative Gaussian curvature. The review is correct in that we do not model a region with positive Gaussian curvature. While this situation should be feasible to create with BUMPY and could be stabilized with dummy particles it does have the drawback of having a single aqueous compartment (see response to point #1 above). We think studying systems with which varying Gaussian curvature is an intriguing idea, but again

something that is beyond the scope of the current study and might be better suited for a publication which is more targeted toward theoretical and physical audiences.

5. The 2D polar coordinate representation seems confusing to me. Why did the authors consider only one (top) flat end and junction region? Shouldn't there be another flat and junction region near $z = 0$? Did the authors consider middle region of the cylinder as $z = 0$? In that case Y axis should be labelled as $|z|$ instead of z .

Response: The latter is correct. We consider middle of cylinder as $z=0$ and are performing two-fold symmetry averaging. We will change the figures to label them with $|z|$. Figs. 3, 5, 6, and S7 have been modified accordingly.

Reviewer #2 (Remarks to the Author):

This study investigates curvature-based lipid sorting in the inner mitochondrial membrane (IMM) using CG Martini lipid force field. The authors used their previously developed approach (JCTC 2018) to build a mitochondrial membrane, in which bilayers are bracketed on both sides by a layer of dummy atoms, which seem to serve as a flat bottom harmonic restraint to prevent the bilayer deviates from the desired geometry. They then tested the impact of dummy atoms on the curved system by evaluating the forces on dummy atoms to ensure the least perturbation on the lipid dynamics.

The lipids simulated were POPC, POPE, DOPE, and cardiolipin (CDL). They found POPE/DOPE and CDL localize in the higher negative curvature regions, consistent with the intrinsic negative curvature of PE and CDL. Correlation analysis shows the lipids are more sensitive to the mean curvature, but also, to a less extent, Gaussian curvature.

Overall, this simulation system is at a biologically relevant length scale and the analysis is carefully done. It is also quite interesting to see that the charge on CDL has minimal effect on lipid sorting as the curvature energetics are the dominant force in the current study. The main limitation of the current study is that only a single IMM shape and size were simulated. It is likely that some variation exists in biology. Comparison across several systems, for example, with variation in the radius of the cylinder or curvature of the junction will significantly improve the current manuscript. Other factors such as changes in lipid rigidity may also affect curvature sorting, which could be potentially investigated here and make the conclusion more comprehensive.

Response: Thank you very much for your time in reviewing the manuscripts and providing us with your comments and suggestions. While these are all excellent suggestions, we believe further investigation into specifics of the geometry and curvature radii would warrant an independent study, as would expanding the number of lipid species under consideration. However, to some degree we have addressed the point about lipid rigidity through our comparison of POPE (16:0/18:1) and DOPE (di-18:1) to understand the effect of increased double bounds in the acyl chains. We believe that the present studied IMM dimensions are biologically relevant and provide a first study at this scale. Certainly in the future we would like to expand the present study calculations with different setup conditions as advised.

Reviewers' comments:

Reviewer #1 (Remarks to the Author):

The revised manuscript has addressed all concerns raised by the reviewers. I recommend the manuscript for publication.

Reviewer #2 (Remarks to the Author):

The major concern that was pointed out by both reviewers is to compare the lipid sorting under different curvatures or slightly different geometry (e.g., the radius of the cylinder). Considering the distribution of Martini lipids converge fast and the geometry of the membrane model is fairly rigid, this reviewer does not believe such simulation is beyond the scope and would significantly improve the manuscript for Commun Bio readers.

Reviewers' comments:

Reviewer #1 (Remarks to the Author):

The revised manuscript has addressed all concerns raised by the reviewers. I recommend the manuscript for publication.

Response: Thank you!

Reviewer #2 (Remarks to the Author):

The major concern that was pointed out by both reviewers is to compare the lipid sorting under different curvatures or slightly different geometry (e.g., the radius of the cylinder). Considering the distribution of Martini lipids converge fast and the geometry of the membrane model is fairly rigid, this reviewer does not believe such simulation is beyond the scope and would significantly improve the manuscript for Commun Bio readers.

Response: We have now included two additional systems. For the POPC/POPE/CDL-2 system, which we believe to be the biologically most relevant, we have created systems in which the cylinder radius was either increased to 15 nm or decreased to 5 nm. Tables 1-3 are updated to include these systems and the results are presented in Figures 10, S16, S17, S21, S22. Figure 11 is updated to include these systems in the correlation analyses. The results show that tighter curvatures do drive more significant partitioning of CDL-2 over POPE, further supporting our main findings.

REVIEWERS' COMMENTS:

Reviewer #3 (Remarks to the Author):

The authors have addressed all of my concerns in this revision. The additional simulation work conducted is of great value and highly appreciated.